# Saline aqueous fluid circulation in mantle wedge inferred from olivine wetting properties

Yongsheng Huang [1*], Takayuki Nakatani [1], Michihiko Nakamura [1] & Catherine McCammon [2]

Recently, high electrical conductors have been detected beneath some fore-arcs and are believed to store voluminous slab-derived fluids. This implies that the for-arc mantle wedge is permeable for aqueous fluids. Here, we precisely determine the dihedral (wetting) angle in an olivine–NaCl–$H_2O$ system at fore-arc mantle conditions to assess the effect of salinity of subduction-zone fluids on the fluid connectivity. We find that NaCl significantly decreases the dihedral angle to below 60° in all investigated conditions at concentrations above 5 wt% and, importantly, even at 1 wt% at 2 GPa. Our results show that slab-released fluid forms an interconnected network at relatively shallow depths of ~80 km and can partly reach the fore-arc crust without causing wet-melting and serpentinization of the mantle. Fluid transport through this permeable window of mantle wedge accounts for the location of the high electrical conductivity anomalies detected in fore-arc regions.

[1] Department of Earth Science, Graduate School of Science, Tohoku University, Aramaki-Aza-Aoba, Aoba-ku, Sendai, Miyagi 980-8578, Japan. [2] Bayerisches Geoinstitut, University of Bayreuth, 95440 Bayreuth, Germany. *email: huang.yongsheng.s8@dc.tohoku.ac.jp

An understanding of fluid circulation in subduction zones is crucial for determining the origin of arc volcanism and to constrain global material recycling[1–5]. Water bounded in the hydrous minerals in the altered subducting slab is continuously released into the overriding mantle wedge via metamorphic dehydration reactions with increasing pressure and temperature during subduction[4,6,7]. The released aqueous fluid can control the partial melting of the mantle wedge because the presence of aqueous fluid effectively decreases the peridotite solidus temperature, which leads to the formation of arc magma[8–11]. The slab-derived water involved in the arc magma returns to the earth's surface via volcanic emission, which is assumed to regulate water cycling in the subduction zone[12].

Water mass balance and geophysical observations, however, have indicated that a water output flux other than volcanic emission should exist in subduction zones. Parai and Mukhopadhyday[12] calculated the global water mass balance constrained by a phanerozoic sea level decrease of 0–360 m and obtained a global water input flux of $1.03–1.17 \times 10^{12}$ kg/yr at the trench. They considered that the output flux in subduction zones is limited through arc/back-arc hydrous magma production. Recently, Faccenda[13] proposed a higher input flux of $1.19–2.0 \times 10^{12}$ kg/yr based on the geophysical constraints on the extent of hydration in the oceanic lithosphere, which suggests the missing output fluid flux in subduction zones to balance the larger amount of water subducted. Although they invoked serpentinized forearc mantle as the water reservoir, analysis of global seismic data revealed that extensive forearc serpentinization is limited to the hottest subduction zone, such as Cascadia[14]. Recent magnetotelluric studies have found moderate-to-high electrical conductivity anomalies at the lower crust to uppermost mantle depths at a nearly constant distance of 20–40 km seaward from the arc volcanoes in the fore-arcs of the Cascadia[15,16], Mexico[17], Costa Rica[18], North Chile[19–21], North Honshu[22], Southern Kyushu[23], and Ryukyu–Philippine subduction zones[24]. Because aqueous fluid effectively increases the bulk electrical conductivity by up to several orders of magnitude[25], these anomalies are interpreted to be reservoirs of aqueous fluid supplied from the subducting slab without causing mantle melting and thus may account for some of the missing output flux in subduction zones. This interpretation requires the presence of a permeable mantle wedge at a relatively shallow depth of ~80 km to enable upward fluid migration.

At high pressure and temperature conditions, fluid migration is controlled by the dihedral (wetting) angle ($\theta$), which is defined by two intersecting interfaces of a pore at a junction with two solid grains. In the equilibrium state, $\theta$ can be expressed as a function of interfacial energy as follows:

$$2\cos(\theta/2) = \gamma_{ss}/\gamma_{sf}, \qquad (1)$$

where $\gamma_{ss}$ is the solid–solid grain boundary energy per unit area, and $\gamma_{sf}$ is the solid–fluid interfacial energy per unit area[26–28]. In the case of a low fluid fraction, as expected in a deep mantle wedge setting[29], the intergranular fluid forms an interconnected network if $\theta < 60°$, whereas the fluid is isolated at grain corners and edges if $\theta > 60°$[27]. The critical $\theta$ may increase at high fluid fractions, which might occur in local areas such as shear zones. Nevertheless, the critical angle of 60° can provide a fundamental constraint on the dominant regime of fluid transport in the deep mantle wedge[28]. Mibe et al.[30] examined the pressure and temperature dependence of $\theta$ in an olivine–$H_2O$ system to constrain the fluid migration in the mantle wedge. They showed that $\theta$ is >60° at forearc depths of 80–105 km; thus, the slab-derived fluid is transported downward within the down-dragged mantle matrix at the base of the mantle wedge. After reaching a depth of ~105 km, where $\theta$ becomes <60°, the fluid can be released into the

hot mantle wedge through the interconnected network to cause hydrous melting just beneath the arc volcano.

Although many previous experiments assumed that slab-derived fluid is composed of pure $H_2O$, recent geochemical studies have demonstrated that NaCl is an important constituent of subduction zone fluids. Fluid inclusions in a peridotite xenolith from a mantle wedge have shown salinity of $5.1 \pm 1.0$ wt% NaCl equivalent[31]. The chemistry of olivine-hosted melt inclusions from primitive arc basalts[32] is characterized by a high $Cl/H_2O$ ratio, which could be attributed to the involvement of aqueous fluid with 1.0–15.0 wt% NaCl in the arc magma. The slab-derived fluids upwelling in forearc regions have shown a chlorine content up to ~4.0 wt% (~6.6 wt% NaCl equivalent)[33–35]. These salinities, particularly those of >5 wt% NaCl equivalent, could be influenced to some degree by the chromatographic fractionation through the hydration reaction in the mantle wedge after the fluid is released from the subducting slab[36–38]. The original salinity of slab-derived fluid equilibrated with dehydrated subducting rocks is thought to be smaller than those values (0.5–2.0 wt% NaCl[39]). Therefore, the influence of NaCl on $\theta$ between olivine and aqueous fluid should be clarified in a wide range of NaCl concentrations to correctly understand fluid circulation in the mantle wedge. Although Watson and Brenan[28] investigated $\theta$ between olivine and aqueous fluid with 27.5 wt% NaCl at 1 GPa and 1000 °C, further systematic studies are required to reveal the pressure and temperature dependence of $\theta$ in a wide range of NaCl concentrations under mantle wedge conditions.

In this study, we constrain the effect of NaCl on the fluid connectivity in the mantle wedge by experimentally determining $\theta$ in both olivine–$H_2O$ and olivine–$H_2O$–NaCl systems at 1–4 GPa and 800–1100 °C by using a piston cylinder apparatus. The NaCl concentration in aqueous fluid ranged from 1.0 to 27.5 wt% in our experimental systems. On the basis of our data, we clarify the grain-scale fluid connectivity and fluid pathway for forming the electrical conductivity anomalies at lower-crust and uppermost mantle depths at forearc regions.

## Results

**Temperature and pressure dependence of the dihedral angle.** The run products were composed of sintered aggregates of olivine and interstitial pores with no other solid phases (Fig. 1). The grain size was larger at higher temperatures as a result of grain growth, reaching ~90 μm at 1100 °C. The pores are generally surrounded by three or more grains with smoothly curved or faceted mineral–fluid interfaces. The curved interfaces were caused by the attainment of constant mean curvature to minimize the surface energy[27,40], whereas the flat interfaces were affected by crystallographically controlled minimum interfacial energy[41]. We measured only olivine–fluid–olivine $\theta$ defined by the curved–curved interfaces to focus on the angle not affected by the facetted planes. Additional examples showing our selection and measurement processes of $\theta$ are given in Supplementary Figs. 1–3. The median of >200 apparent angles measured on cross-sectional images was adopted as the true $\theta$[42]. Although a true $\theta$ generally occurs in a range owing to the surface energy anisotropies of minerals, the median value can represent the most frequent true $\theta$ in the system. Details on the experimental conditions and results as well as frequency distribution histograms of the measured apparent $\theta$ are given in Supplementary Table 1 and Supplementary Figs. 4, 5, respectively.

The cumulative frequencies of the measured apparent $\theta$ in both $H_2O$ and $H_2O$–NaCl (27.5 wt%) systems along with the median angles are shown in Fig. 2. The same diagrams for $H_2O$–NaCl systems with 1.0, 3.0, 5.0, 10.0, and 15.0 wt% NaCl are shown in Supplementary Fig. 6. The cumulative frequencies always show a

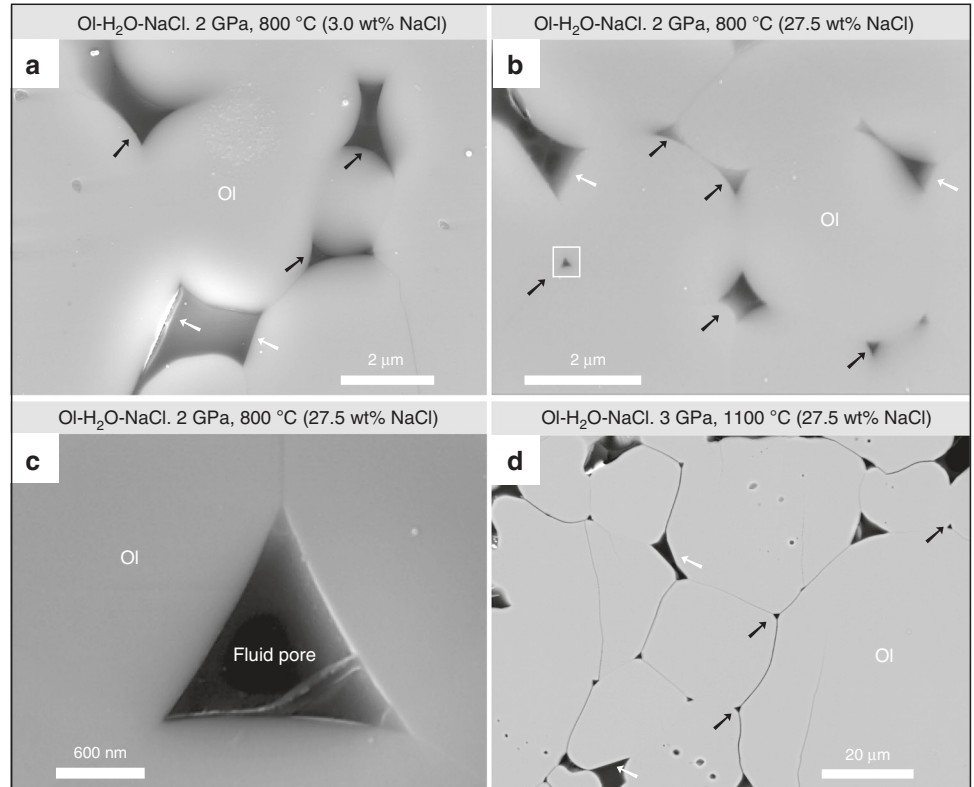

**Fig. 1** Representative scanning electron microscope images. **a**, **b** Secondary electron images of run products with 3.0 and 27.5 wt% NaCl for Runs SCD-17 and CDM-08 at 800 °C and 2 GPa for 192 and 210 h, respectively. **c** High-magnification secondary electron image of the rectangular area shown in **b**. **d** Backscattered electron image of the run product at 3 GPa and 1100 °C for 72 h with 27.5 wt% NaCl (Run CDM-18). The run products are composed of olivine grains (gray) and epoxy resin-filled pores (black) that were previously filled with aqueous fluid during the experiment. The interfaces are generally characterized as smooth and curved (black arrows) but are sometimes associated with flat planes (white arrows).

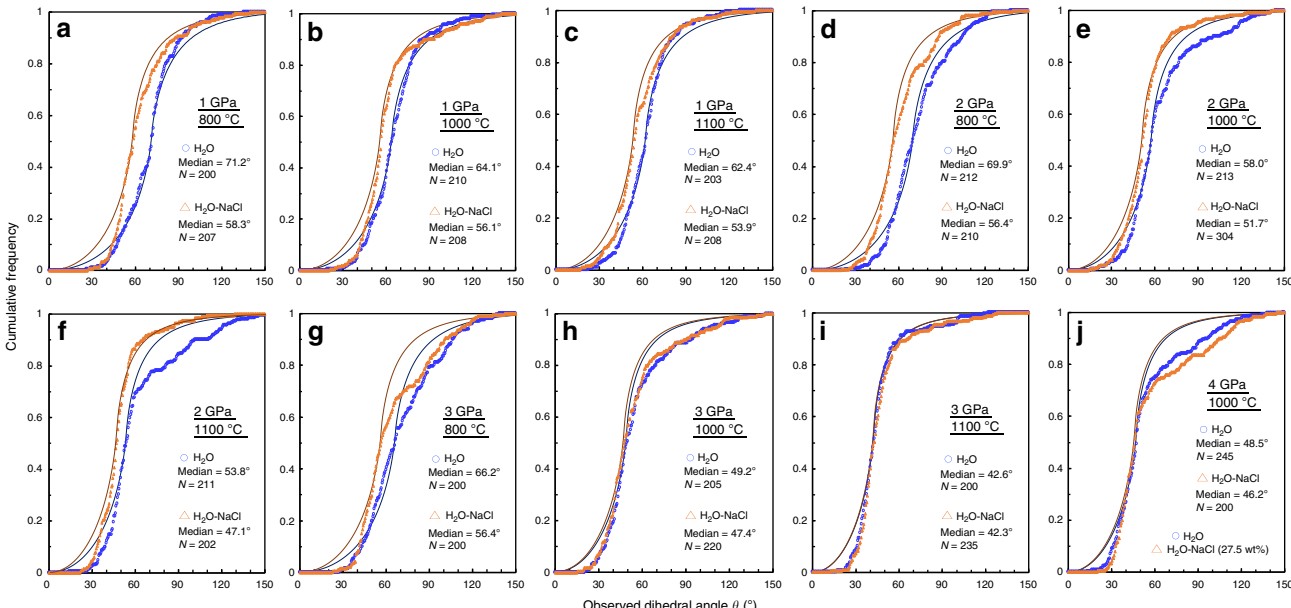

**Fig. 2** Cumulative frequency curves of the measured dihedral angles. **a–j** Blue circles and orange triangles denote the data in the $H_2O$ and $H_2O$–NaCl systems, respectively. The median value and number (N) of measured angles are shown for each experimental condition. The thin lines represent the theoretical cumulative frequency curves of the isotropic system with one true $\theta$, which is assumed to coincide with the obtained median value, where the blue lines represent the olivine–$H_2O$ system, and the orange lines represent the olivine–$H_2O$–NaCl (27.5 wt%) system.

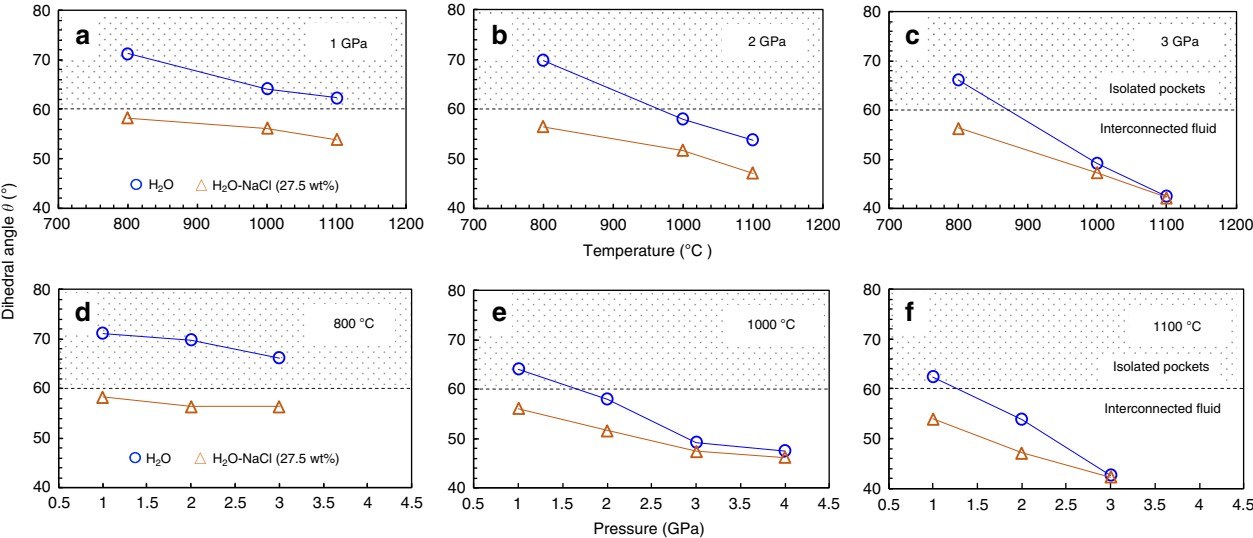

**Fig. 3** Temperature and pressure dependence of dihedral angles. **a–c** Temperature dependence. **d–f** Pressure dependence. The blue circles and orange triangles represent the data in the $H_2O$ and $H_2O$–NaCl (27.5 wt% NaCl) systems, respectively. In the pattern-filled area, $\theta$ is larger than the critical value of 60°, and the intergranular fluid is thus isolated in the fluid pockets; the fluid is interconnected when $\theta$ is <60°.

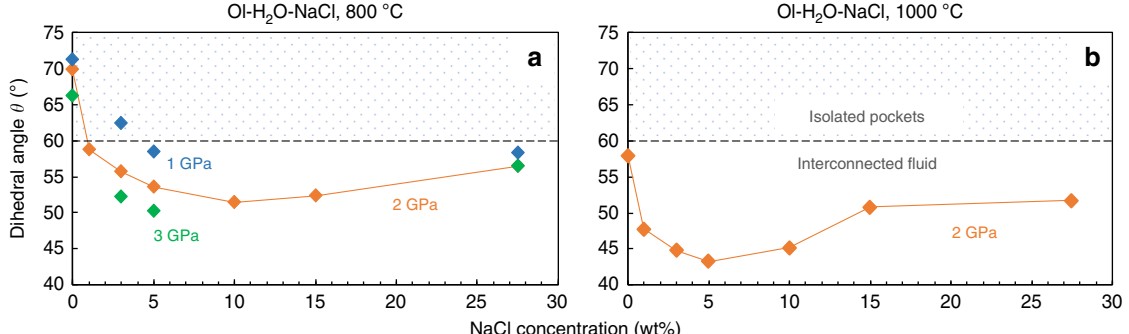

**Fig. 4** Effects of NaCl concentration on the dihedral angle **a** Median $\theta$ at 800 °C and 1–3 GPa. **b** $\theta$ at 1000 °C and 2 GPa. Blue, orange, and green diamonds represent $\theta$ in the olivine–$H_2O$–NaCl systems at pressures of 1, 2, and 3 GPa, respectively. In the pattern-filled area, $\theta$ is larger than the critical value of 60°.

steep increase around the median $\theta$, which is consistent with the theoretical prediction for an isotropic system with one true $\theta$. However, in some experiments, the data largely deviated from the theoretical curve at the region of high apparent $\theta$. This discrepancy can be explained by the presence of a few cases of very large $\theta$ associated possibly with low-angle grain boundaries, where the misorientation between the two adjacent grains is very small[43]. Even in this case, however, the median angle is assumed to represent the system because the distribution of the true $\theta$ is concentrated mostly at the median angle.

The pressure and temperature dependence of the median $\theta$ in both $H_2O$ and $H_2O$–NaCl (27.5 wt% NaCl) systems is shown in Fig. 3. The $\theta$ decreased with increases in pressure and temperature in both systems. Moreover, $\theta$ in the $H_2O$–NaCl system was always <60°, and was also smaller than that in the $H_2O$ system at the same pressure and temperature conditions. The $\theta$ in the $H_2O$ system, however, was <60° only at pressures and temperatures above 2 GPa and 1000 °C, respectively. The difference in $\theta$ between these two systems was up to 13° at 1 GPa and 800 °C, but gradually decreased with increases in pressure and temperature, mostly diminishing above 3 GPa and 1000 °C, respectively.

**Effects of NaCl concentration on the dihedral angle.** The effects of the NaCl concentration on $\theta$ in the olivine–$H_2O$–NaCl system at temperatures of 800 and 1000 °C under pressures of 1–3 GPa are

shown in Fig. 4. At all pressure and temperature conditions, $\theta$ showed a nonlinear dependency on the NaCl concentration in aqueous fluid: The angle steeply deceased at low concentrations and then gradually increased with increasing NaCl concentration. At 2 GPa and 800 °C, for example, $\theta$ suddenly dropped from 69.9° to 58.7° by adding 1 wt% NaCl to the pure $H_2O$ fluid and continued to decrease down to 51.5° with increasing NaCl concentration up to 10 wt%. Further addition of NaCl, in contrast, slightly increased the angle up to 56.4° at 27.5 wt% NaCl. It is worth noting that these angles are still clearly smaller than those in the pure $H_2O$ system (Fig. 4a). With an increase in pressure from 1 to 3 GPa at 800 °C, the angles decreased at all of the investigated concentrations; the decrease was to a higher degree at lower concentrations (Fig. 4a). For example, $\theta$ >60° ($\theta = 62.4°$) at 3 wt% NaCl at 1 GPa became <60° at pressures above 2 GPa. With an increase in temperature from 800 °C to 1000 °C at 2 GPa, the angles became smaller at all of the experimental concentrations; the extent of the reduction was again larger at lower concentrations (Fig. 4a, b). The NaCl concentration at the point at which the angle reached its minimum was lower at 1000 °C and 2 GPa (~5 wt% NaCl) than at 800 °C and 2 GPa (~10 wt% NaCl; Fig. 4a, b).

## Discussion

The median angles in our experiments can be compared with those reported previously at the same pressure and temperature

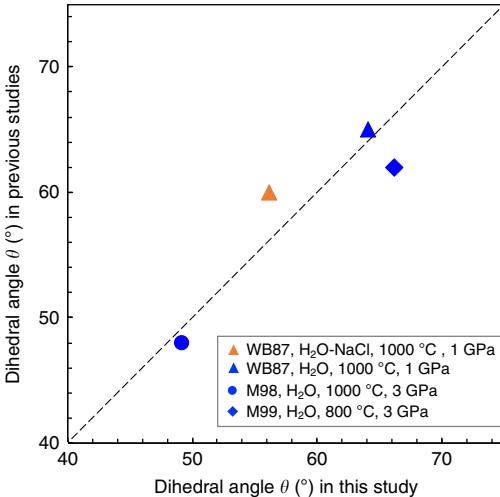

**Fig. 5** Comparison of our results with previous studies In the olivine–$H_2O$ system, the dihedral angles ($\theta$) at 1000 °C were consistent with those reported in previous studies, whereas $\theta$ at 3 GPa and 800 °C was slightly larger than that in Mibe et al.[30]. This discrepancy is attributed to the effect of the faceted plane. In the olivine–$H_2O$–NaCl system (27.5 wt% NaCl), our data are smaller than the results of Watson and Brenan, which was most likely caused by differences in the measurement number.

conditions in the same systems[28,30,44](Fig. 5). In this study, we focused only on the angles defined by the curved–curved interfaces, whereas Watson and Brenan[28] and Mibe et al.[30] measured the angles, including faceted interfaces. Laporte and Provost[45] demonstrated that a median angle with faceted planes became smaller than that without faceted planes, particularly in a system with enhanced crystal anisotropy. Watson and Brenan[28] and Mibe et al.[44] reported $\theta$ values of 65.0° and 48.0° in an olivine–$H_2O$ system at 1000 °C under pressures of 1 and 3 GPa, respectively. These angles are in good agreement with our data at the same pressure and temperature conditions of 64.1° at 1 GPa and 1000 °C and 49.2° at 3 GPa and 1000 °C. This demonstrates the minor effects of crystal anisotropy at 1000 °C on the median angle and good reproducibility of the measurements (Fig. 5). In contrast, at 800 °C and 3 GPa in the $H_2O$ system, the median angle reported by Mibe et al.,[30] at $\theta = 62.0°$, is slightly smaller than that in our experiment, at $\theta = 66.2°$ (Fig. 5). This discrepancy might be indicative of the enhanced contribution of faceted planes on the median angle at 800 °C. Although our measurements do not include the angle with faceted interfaces, its inclusion would not have made a remarkable difference between the two investigated systems, and thus would have not changed our main conclusions.

The median angle in the $H_2O$–NaCl system with 27.5 wt% NaCl at 1 GPa and 1000 °C from Watson and Brenan[28], at $\theta = 60.0°$, is slightly higher than that in our experiment, at $\theta = 56.1°$, which cannot be explained by the difference in the type of measured angle. In order to check the reproducibility in a different laboratory, we conducted one piston cylinder experiment in the $H_2O$–NaCl system at the same pressure and temperature conditions at Tohoku University and obtained almost the same median angle (Run CDMR-04) as that obtained at Bayreuth University (Run CDM-04), at $\theta = 55.8°$ and 56.1°, respectively. Because the number of measured angles in this study, at 208, is significantly larger than that in Watson and Brenan[28], at 100 ± 15, we are convinced that our precise measurements are more reliable than those in the previous study.

The observed decreases in $\theta$ with the addition of NaCl is indicative of reduced interfacial energy between the olivine and

aqueous fluid in the $H_2O$–NaCl system under the assumption that the grain boundary energy between two olivine grains is constant. Takei and Shimizu[46] theoretically demonstrated that the interfacial energy between the solid and fluid phases decreases with an increase in solubility of the solid component into the fluid phase in binary eutectic systems, such as silicate+ fluid, binary alloys, and binary organic systems. In an olivine–$H_2O$ system, the experiments conducted by Yoshino et al.[47] showed that $\theta$ decreased down to 0° with an increase in pressure up to ~7 GPa owing to the enhanced solubility of olivine into the aqueous fluid. Because NaCl is known to increase the forsterite solubility into the aqueous fluid at 1 GPa[48], we consider that high olivine solubility effectively reduces the interfacial energy and thus $\theta$ between olivine and aqueous fluid in an $H_2O$–NaCl system. As shown in Fig. 3, essentially no difference was detected in $\theta$ between the $H_2O$ and $H_2O$–NaCl systems at pressures and temperatures >3 GPa and 1000 °C, respectively. This fact suggests that the NaCl effect on olivine solubility became less remarkable at such high pressure and temperature conditions owing possibly to the elevated solubility of olivine in the pure $H_2O$ system. At low NaCl concentrations, $\theta$ significantly decreased, reached its minimum values at 5–10 wt%, and then gradually increased. This is consistent with the data showing that olivine solubility steeply increases at low NaCl concentrations, and is suppressed at very high concentrations[48]. The interfacial energy between minerals and fluids decreases by positive adsorption of the solute on the mineral surface[49]. This could also contribute to the nonlinear dependence of $\theta$ on the NaCl concentration (Fig. 4).

The dramatic effects of NaCl in decreasing $\theta$ in the olivine and aqueous fluid system led us to apply an alternative fluid circulation model in the subduction zones. Here, we assume that the grain-scale fluid connectivity in the mantle wedge is controlled by $\theta$ between olivine and aqueous fluids, because olivine is the dominant mineral phase in the upper mantle. Although pyroxenes in the mantle could have larger solid–fluid interfacial energy than olivine, their influence on the fluid connectivity could be minor in the olivine-dominated peridotite system[50].

The 60° isopleths of $\theta$ in both the $H_2O$ and $H_2O$–NaCl systems, which are the thresholds for the formation of an interconnected fluid network in the mantle, are shown in a depth–temperature diagram in Fig. 6. We also drew the 60° isopleths for low NaCl concentrations of 1.0, 3.0, and 5.0 wt%. Although the measured $\theta$ showed a minimum value around medium NaCl concentrations of 5.0–10.0 wt% (Fig. 4), the low NaCl concentrations, at 1.0–5.0 wt%, had a strong effect on the decrease in $\theta$. These NaCl concentrations are assumed to cover the salinities of slab-derived fluids (0.5–2.0 wt%[39]; 5.1 ± 1.0 wt%[31]; 1.0–15.0 wt%[32]; 6.6 wt%[33–35]). Details on the isopleths of $\theta$ other than 60° are provided in Supplementary Figs. 7–9. Also shown in the figure are the depth–temperature paths of the base of down-dragged mantle wedges in intermediate-temperature and cold subduction zones (D80 models of Syracuse et al.,[51] and van Keken et al.[4]), where high electrical conductivity has been detected in deep forearc regions of Costa Rica[18], North Chile[19–21], North Honshu[22], Southern Kyushu[23], and Ryukyu-Philippines[24]. In these thermal models, rheological coupling between the subducting slabs and hot mantle wedges is assumed to occur at a nearly constant depth of 80 km, which leads to dramatic changes in the thermal gradient around the coupling depth. The intersection depth between the 60° isopleth and the pressure and temperature paths of the basal layer of the down-dragged mantle wedges in the $H_2O$–NaCl system (~80 km) is obviously much shallower than that in the $H_2O$ system (>100 km) even at a low NaCl concentration of 1.0 wt% (Fig. 6). This implies that the NaCl-bearing aqueous fluid can form a grain-scale interconnected network in the shallow mantle wedge, enabling pervasive fluid migration at

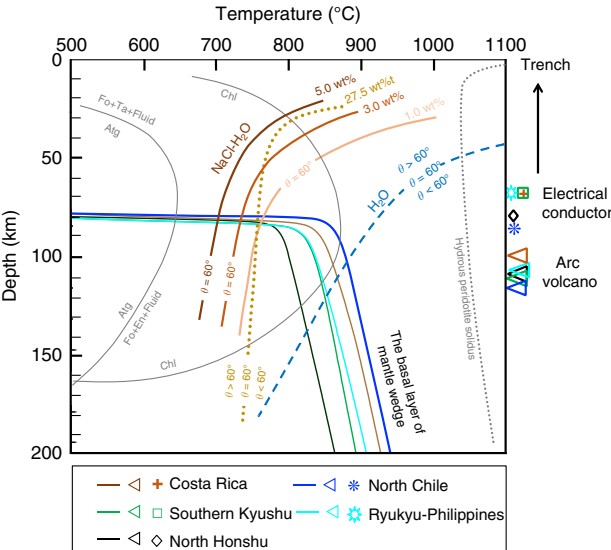

**Fig. 6** 60° isopleths of dihedral angle in depth–temperature space The thick brown, red–orange, and orange curves are the 60° isopleths in the low-NaCl-concentration systems; the yellowish-brown dotted curve is the 60° isopleth for the 27.5 wt% NaCl; and the blue dashed curve is the 60° isopleth in the $H_2O$ system. In pressure and temperature regions higher than the 60° isopleths, $\theta$ is smaller than 60°. The depth–temperature paths of down-dragged mantle at the base of mantle wedge are denoted by five thin solid lines of different colors for intermediate-temperature and cold subduction zones[4,51], where high electrical conductivity has been detected in the deep forearc crust. The slab-surface depths beneath the forearc conductor and the arc volcano are shown at the right side of the panel for each subduction zone[77]. The gray dotted line shows the hydrous peridotite solidus that is after Green[67]. It should be noted that this hydrous peridotite solidus was constrained in a fertile peridotite system; the solidus temperature in a relatively depleted peridotite should be higher than that shown in this figure. The stability of antigorite is after Bromiley and Pawley[78] and Evans et al.[79], and that stability of chlorite is after Till et al.[11]. Atg antigorite, Fo forsterite, Ta talc, En enstatite, Chl chlorite.

depths of ~80 km. Although shear deformation is expected at the base of the flowing mantle wedge, such deformation does not significantly affect the fluid connectivity because textural equilibrium is reached much faster than creep deformation, particularly at elevated pressure and temperature conditions[52].

In cold and intermediate-temperature subduction zones, eclogite transformation of the subducting oceanic crust, which feeds voluminous amounts of aqueous fluid into the mantle wedge, is assumed to begin at depths of ~75–80 km[53]. This dehydration continues to depths of 90–140 km depending on the plate age and subduction velocity[53]. Therefore, the saline fluid liberated from the subducting slab can penetrate into the permeable mantle wedge from the beginning of the eclogite transformation. The pressure and temperature conditions of the mantle wedge where such slab-derived fluids are supplied is beyond the stability field of serpentine in these subduction zones. In our model, we excluded hot subduction zones such as Cascadia and Mexico, where major slab dehydration occurs at very shallow depths of 40–60 km and causes the fluid to circulate though the stagnant and extensively serpentinized forearc mantle wedge[54].

Fluid migration mode other than grain-scale pervasive infiltration may also be possible at depths at which the saline fluid forms an interconnected network in the mantle wedge. Porosity waves driven by matrix compaction is regarded as an efficient mode of fluid transport in a viscoplastic environment[55,56], which can occur through a fluid network in the matrix when the pore

fluid pressure reaches a certain value. Localized flow along cracks or shear zones can be effective pathways for transferring aqueous fluid especially in the lower crust and the adjacent uppermost mantle. Nevertheless, the grain-scale pervasive flow is an important mode of fluid migration that can prevail in a sub-solidus mantle wedge.

To infer the consequences of shallow infiltration of NaCl-bearing aqueous fluid, the slab-surface depths beneath the forearc electrical conductors and the arc volcanic fronts are shown in Fig. 6 for each subduction zone. In the olivine–$H_2O$ system, the 60° isopleth intersects with the depth–temperature paths of down-dragged mantle at depths of 100–140 km, which roughly correspond to the positions of the arc volcanoes. Mibe et al.[30] argued that slab-derived fluid released at depths of 80–105 km is trapped in the down-dragged mantle wedge owing to the large $\theta$ and that these fluids are suddenly released to the hot mantle wedge core at depths of ~105 km, where $\theta$ becomes <60°. This causes hydrous mantle melting just beneath the arc volcano. However, if we consider the effect of fluid salinity on $\theta$, the model of the pure water system appears to be questionable. Our model shows that the 60° isopleth intersects with the depth–temperature paths at depths shallower than ~80 km irrespective of the NaCl concentration (i.e., 1.0 to 27.5 wt% NaCl) owing to the steep thermal gradient at the coupling depth of 80 km, which is inconsistent with the depth of the volcanic front at ~100 km. Strikingly, the depth of the intersection is strongly consistent with the position of the high electrical conductor in the deep forearc crust. The coupling depth is constrained within a narrow range of 70 to 80 km by observations of surface heat flow and seismic attenuation in the forearc region[53,57]. Therefore, the relationship between the coupling depth (i.e., depth of fluid interconnection) and the slab-surface depths beneath the forearc electrical conductors is robust.

The above relationship can be explained if we assume that the NaCl-bearing fluid released at the beginning of the dehydration reaction in subducting slab percolates upward through the interconnected network in the mantle wedge and accumulates at crustal depths as a fluid reservoir (Figs. 6, 7), because the saline fluid has very high electrical conductivity[25,58–60]. Although partial melting and interconnection of conductive minerals are also invoked to explain the high conductivity[61–64], the thermal gradient in the forearc region is too low to cause melting, and the volume of such conductive minerals is generally too small to form interconnected network. Recent high-resolution magnetotelluric studies have found a continuous and moderately conductive anomaly in the mantle wedge rooted from the top of the subducting slab to the forearc crust[18,21,65]. These observations can support the steady-state fluid transport via intergranular fluid networks in the mantle wedge with a caveat of a smoothing artifact. While such a continuous conductive anomaly was not always observed in other studies, the interconnection at small fluid fraction in the mantle wedge is still not denied due to the fluid fraction dependence of the electrical conductivity[25] and the electrical static effect of the shallow high electrical conductor on electrical amplitude response of deep electrical field[66]. Electrical conductivity in the deep forearc crust could be enhanced by the high fluid fraction due to the fluid accumulation over the geological time scale and the high fluid salinity resulted from the hydration of the mantle wedge during fluid transport.

At higher temperatures, aqueous fluids significantly decrease the solidus temperature of peridotite. If mantle is partially molten by fluid fluxing, the volatile transport and physical properties of rocks are determined by incipient partial melts. Till et al.[11] conducted melting experiments in a hydrous fertile peridotite system, and proposed that the incipient melting of mantle wedge occurs at 890–870 °C in the chlorite-bearing water-

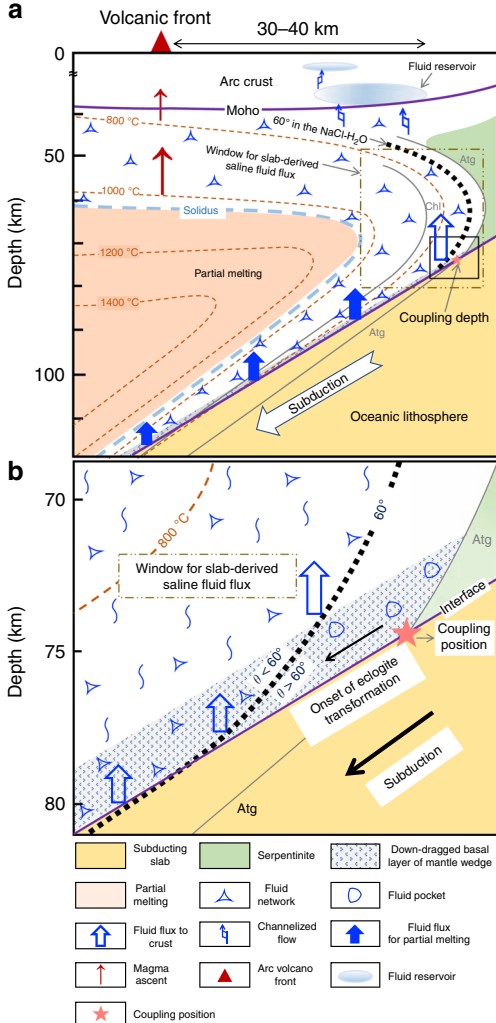

**Fig. 7** Schematic model for migration of saline aqueous fluids. **a** Overview of subduction fluid migration. **b** Enlargement of the black rectangle in **a**. Slab-derived saline fluids can form an interconnected network in the mantle wedge at depths of ~80 km, and can percolate partly through the overriding mantle without causing melting or serpentinization to form a fluid reservoir in the forearc crust (blue thick open arrow). At depths <~80 km, where antigorite is still stable, slab-derived fluids can enter the cold corner of mantle wedge as a channelized flow along cracks. At depths >~80 km, saline fluid continuously infiltrates the mantle wedge with temperatures above ~1050 °C, which triggers the partial melting of peridotite[67] (blue thick solid arrows). It should be noted that this hydrous peridotite solidus was constrained in a fertile peridotite system; the solidus temperature in a relatively depleted peridotite should be higher than that shown here. Magma ascent results in the formation of arc volcanoes (red arrows), although the location of the volcanic front is not directly related to the dihedral angle threshold for saline fluid. The geothermal structure is after Wada et al.[80]; the stability of antigorite is after Bromiley and Pawley[78] and Evans et al.[79]; and the stability of chlorite is after Till et al.[11]. Atg antigorite, Chl chlorite.

undersaturated system and 850–820 °C in the water-saturated system at pressures of 2–3 GPa. However, these solidi were questioned by the recent research which has found that solidus of water-saturated fertile peridotite is ~1000 °C at 1–3 GPa[67]. Although no precise experimental data are available for depleted peridotite, which is more applicable to common mantle wedges, it should be higher than that for the fertile peridotite. At lower temperatures, however, serpentine is formed by water infiltration;

thus, the slab fluid may be sequestered until the peridotites are dominantly transformed to serpentinites[54]. It should be emphasized that the temperature gap exists between the hydrous peridotite solidus and serpentine stability, which serves as a window for slab-derived fluid to percolate completely through the mantle wedge and accumulate to form the fluid reservoir near the Moho (Fig. 7a). This fluid reservoir may account for some part of missing output fluid fluxes in subduction zones in addition to the high electrical conductivity anomalies detected beneath the forearc regions.

The stability of chlorite expands at much higher pressures and temperatures than those of serpentine, at a maximum of 870 °C at 2–3 GPa[11]. Slab-derived aqueous fluid is expected to be consumed by the formation of chlorite in the mantle wedge, indicating that sustainable supply of fluid is required to maintain a water-saturated condition. The subducting oceanic lithosphere can provide such a water source by dehydration reactions. Chlorite is likely to form along grain boundaries between nominally anhydrous minerals, such as olivine, orthopyroxene, clinopyroxene, and spinel. Thus, the wetting behavior seems to be affected by chlorite, which is supposed to be influenced strongly by the faceting of mineral surfaces. However, the influence of chlorite in fluid connectivity at the forearc mantle wedge is most likely limited by the modal composition of chlorite and its stability. First, the modal composition of chlorite formed in peridotite systems is limited by the aluminum content in the system, and thus reaches a maximum of 10–15 vol % in the depleted lherzolite and harzburgite[68]. Second, as NaCl in slab-derived fluids can reduce the water activity effectively, the stability limit of chlorite would shift to the lower temperatures. Thus, chlorite would be unstable in the P–T conditions investigated by this study, and grain boundaries in peridotite are dominantly composed of olivine. Therefore, the formation of minor amount of chlorite could not significantly affect the fluid transport. Furthermore, chlorite formation may increase the salinity of the aqueous fluid[36], which can further reduce the stability of chlorite itself and enhance the electrical conductivity of the slab-derived fluid.

Saline fluid is considered to be an effective agent for transportation of subduction components such as Rb, Sr, La, Ce, and U in arc magmas[69]; however, its quantitative evaluation is still under debate. The contribution of hydrous silicate melt derived from the melting of subducted oceanic crust and sediments has also been invoked to explain the trace element chemistry of arc magmas[70,71]. Such melting, however, occurs at pressures higher than 3.5 GPa, and thus does not affect the transport of aqueous fluid beneath high electrical conductors in forearc areas.

In conclusion, the dihedral angle in the olivine–H$_2$O–NaCl system is small in the wide pressure–temperature and fluid salinity ranges, establishing the interconnected fluid network. The transportation of slab-derived saline aqueous fluid occurs in between the serpentine stability field and wet-melting region most possibly along such a grain-scale fluid network and thus account for the high electrical conductivity anomalies detected in the forearc regions.

## Methods

**Experimental procedures**. Annealing experiments were conducted in olivine–H$_2$O and olivine–H$_2$O–NaCl systems to examine the effect of NaCl on dihedral (wetting) angle ($\theta$) between olivine and aqueous fluid. The starting material was San Carlos olivine (Fo$_{91}$Fa$_9$) handpicked from a gently crushed San Carlos lherzolite xenolith. These grains were finely ground with a tungsten carbide mortar, and were sieved to obtain size fractions of 38–53 μm. Impurities included in the powder were carefully removed under an optical microscope. To prevent absorption of moisture, the olivine powder was placed in an oven, and was held at 100 °C for one night prior to the experiment. Deionized and distilled water was used for the source of pure H$_2$O. We dissolved reagent-grade NaCl (99.99% NaCl)

into the deionized and distilled water at room temperature (~25 °C) and atmospheric pressure to obtain solutions with various NaCl concentrations of 1.0, 3.0, 5.0, 10.0, 15.0, and 27.5 wt% NaCl.

End-loaded 3/4- and 1/2-inch piston cylinder apparatuses at Bayerisches Geoinstitut, University of Bayreuth, were utilized for the experiments at pressures of 1–2 GPa and 3–4 GPa, respectively, with a standard talc–Pyrex assembly[72]. The olivine powder along with ~3–5 wt% fluid was loaded in an end-welded noble metal capsule that was sealed by arc welding. Capsules constructed of Au and $Au_{80}Pd_{20}$ alloy with outer diameters of 2.2 mm and 2.0 mm, respectively, were used for the experiments at temperatures of 800–1000 °C and 1100 °C, respectively. The cells were first pressurized to 90% of the target pressure; the remaining 10% pressure was applied after heating to the target temperature. During the experiments, the temperature was monitored and controlled by using an S-type thermocouple ($Pt$–$Pt_{90}Rh_{10}$). The run durations varied from 72 to 210 h, depending on the target temperature. Quenching to room temperature was completed within 1 min, and the pressure was slowly released until atmospheric pressure was reached.

To check the reproducibility of our measurement, we conducted one annealing experiment at 1 GPa and 1000 °C in the $H_2O$–NaCl system for 120 h by using the end-loaded 3/4-inch piston cylinder apparatus at Tohoku University. A conventional NaCl–Pyrex pressure assembly was utilized for the experiment. We used the same starting materials as those used in the experiments at Bayerisches Geoinstitut. The starting materials were packed into an Au-lined machinable ceramics cylinder (MACOR, $MgO + Al_2O_3 +$ F-micas), which was also used in the experiments by Jégo et al.[73]. A thin Au disk was then placed on top of the Au capsule, and the Au capsule inside the cylinder was shut following the initial pressurization of the run. The temperature gradient along the capsule was estimated to be <10 °C at 1200 °C, according to Nakamura and Watson[74]. The detailed procedures for these high-pressure experiments are similar to those described in Ohuchi and Nakamura[75].

**Sample evaluation**. The recovered capsules were cut with a wire saw to expose the run products. The run products were then impregnated with epoxy under a vacuum and were polished down to 1.0 µm with alumina powder and subsequently to 0.06 µm by colloidal silica suspension. The polished sections of the run products were observed by using a field emission-type scanning electron microscope. More than 200 backscattered electron (BSE) or secondary electron (SE) images of 1280 × 960 pixels were obtained for each run product at magnifications of ×3000 to ×150,000, depending on the pore size. The $\theta$ measurements were performed by using Image-J software for >200 angles on the acquired SE images for each sample. Errors of each measurement were <3°. The details for discussing error are provided in Supplementary Note 2. The true $\theta$ was estimated by calculating the median of the distribution of apparent angles[42,76]. This number of measurements was necessary to obtain a steady median value, particularly when the sample incurred severe plucking, which can locally reduce the randomness of the apparent angle distribution. Additional examples showing the selection and measurement of $\theta$ are given in Supplementary Note 1 and Supplementary Figs. 1–3.

## Data availability
All data used in this study are available in the Supplementary Data published alongside this manuscript. In addition, the data that support the findings of this study are available from the corresponding author upon reasonable request.

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

## Acknowledgements

We are grateful to Haihao Guo, Masahiro Ichiki for their technical support and constructive suggestions. This work was supported by JSPS KAKENHI Grant Nos. JP16H06348 and JP16K13903 awarded to M. Nakamura, JSPS Japanese–German Graduate Externship, International Joint Graduate Program in Earth and Environmental Sciences, Tohoku University (GP-EES), and by the Ministry of Education, Culture, Sports, Science and Technology (MEXT) of Japan under its Earthquake and Volcano Hazards Observation and Research Program, and by the Core Research Cluster of Disaster Science in Tohoku University (Designated National University).

## Author contributions

All authors contributed to the preparation and revision of the paper. Y.H. conducted the experiments, measured the dihedral angle, and determined the implications. T.N. calculated the theoretical cumulative frequency curve of the dihedral angle and contributed to the application of the experimental results to the fluid circulation model. M.N. conceptualized the basic experimental design and contributed to interpreting the results. C.M. provided the specific direction of experiment.

## Competing interests

The authors declare no competing interests.
