## [Peer Review File · Nature Communications]

Reviewers' Comments:

Reviewer #1:

Remarks to the Author:

This is a very interesting and a well-written paper that was a pleasure to read. The experiments are carefully conducted. There is a good range of experiments and careful evaluation of the textures, with a large number of observations. The main result is that the dihedral angle decreases from a pure water-olivine system to a water+salt-olivine system. This conclusion appears to be solid and it is used to infer fluid transfer in static mantle peridotites. However, I have some concerns on how these experimental results are then up-scaled to explain main features of water recycling in subduction zones.

Main comments referring to line Nr in paper:

1) Lines 4, 29-30, 36-37 suggest that a hidden reservoir is needed to balance the inputs and outputs in subduction zones. This is in conflict to the correct statement in lines 22-24, that there is a continuous release of water from the subducted crust during prograde subduction metamorphism. We are far away from fully quantifying how much water is recycled from the slab via fore-arc mantle or accretionary prisms to the crust. It seems a problem is created here to offer a solution that might not be needed.

2) Lines 53-54. The dihedral angle of the olivine fluid-system is controlling fluid flow at low fluid fractions. All experiments were performed at low fluid contents of 3-5 wt%. However, it is not clear if this is the case when a fluid enters the mantle wedge. If fluids are collected in shear zones in the subducting slabs, it is very likely that the fluid fraction is high and in such case the fluid is expected to be mobile. The generalisations are not always justified.

3) Lines 62-70: This is quite a good summary of evidence for saline fluids in subduction zones. The very high NaCl contents in Alpine serpentinite have to be taken with care as they most likely derive from a desiccation process (see also point). Overall most evidence points to salinities < 5 wt% NaCl. The main investigated conditions with 27.5 % NaCl is really on the high side for aqueous subduction zone fluids. I acknowledge that two salinity series were done at constant conditions, indicating that the results are also valid at lower salinities, but it would have been nice to have the bulk of the data at more realistic salinities.

4) 215-216 The position of the second critical point is highly debated and the conditions at 3.8 GPa, 1000°C refer to a fertile peridotite+H₂O system. However in the current study deals with brine-olivine and the second critical point for this system is nowhere near the investigated P-T conditions but at much higher pressures and temperatures (constrained by the wet melting curve of Mg-rich olivine). Thus, this argument cannot be taken to explain the experimental results.

5) 249-252 Fluid production in subducted mafic crust is more complex and most eclogites are dry at 650-700°C, 25 kbar. This doesn't really fit the model as at these conditions, the dihedral angle is still > 60° even for the saline fluids. Thus, fluid transport in the mantle wedge according to the presented arguments is not likely and much of this fluid might escape along the slab interface. Alternatively, especially at mantle wedge temperatures < 650°C serpentine minerals will form due to fluid infiltration. It would be good to also show the stability of antigorite in Figure 6. This would highlight that fluids derived from mafic rocks would mainly be produced in a region, where antigorite is still stable in the mantle wedge.

6) Lines 271-77; Figure 7 284-288. As noted above, the dehydration of oceanic crust is not correctly shown in this figure, as most of the oceanic crust at the top of the slab will be anhydrous before the coupling point is reached. It is also important to note that in Figure 6 there is a gap between the stability of antigorite in the mantle wedge and the line of $\Theta = 60^\circ$. This gap somehow disappeared in Figure 7. However, this is quite important for how fluid is transported in this critical region in the mantle wedge.

7) 291-294. Here it is assumed that the saline fluid is produced directly in the subducted slab. This is not easy to do. The Cl/H₂O ratio of dehydrating phases is generally much lower than salinities of 5 wt% (see for example Li and Hermann, 2015). One proposed mechanism that has been used to also explain the electric conductivity is that saline fluids are not derived from the slab but are generated by the formation of hydrous phases such as serpentine and chlorite in the mantle wedge

where fluids are getting enriched in Cl by desiccation (see for example Reynard et al., 2011, EPSL)

8) Throughout the discussion (280-284), the authors give the impression that arc lavas are just formed by the addition of water to the mantle wedge. While water is the key ingredient, all arc lavas are also characterised by a "slab component" that contains major and trace elements. With the proposed model, the main subduction fluids are released from the slab at quite low temperatures and pressures and then they travel to hotter parts trapped within the down-dragged coupled mantle. At 500-600°C, even in highly saline fluids, the solubility of LREE is far too low to explain the H₂O/Ce ratios in arc lavas (see for example Plank et al., 2009). Thus, the proposed model might partially work for water but it seems difficult to reconcile it with additional constraints on the composition of subduction zone fluids.

Additional General comments:

9) Only static conditions at low fluid fractions are investigated. However, the subduction channel and mantle wedge are dynamic environments and thus fluid flow in a deforming system might be quite different. This point needs to be addressed.

10) It has been proposed based on different interaction features of slab fluids with the olivine that fluid flow in the mantle wedge might be channelized (see for example Pirard and Hermann, 2015 Geology).

11) Porosity waves as advocated by Connolly and Podladchikov are not considered as a potential mechanism to transport fluids through the mantle wedge. This must be addressed as well.

12) I recommend limitation of the application of the results to the conductivity of fluids within a more static low T mantle wedge, rather than trying to explain the whole subduction-recycling system. Especially in the region where chlorite-peridotite is stable in the fore-arc region, the model could be nicely applied as the formation of chlorite would provide a means to create high salinity fluids, and the dihedral angle with olivine could explain then the migration of these fluids through the fore-arc, providing an explanation for the electrical conductivity and magnetotelluric studies.

Figure 6 is very schematic and hard to read. The depth temperature paths of the subducted slabs are very systematic. It is highly unlikely that all subduction zones will couple to the mantle at exactly the same depth. It would be good to show also the antigorite stability in this diagram. The wet peridotite solidus is not accurate as well, especially at the low pressure and high pressure end of the solidus.

Figure 7 does only partly reflect the data in Figure 6 (see point 6).

Bern, 6.12.2018

Jörg Hermann

Reviewer #2:

Remarks to the Author:

The paper reports a variation of dihedral angle of olivine-NaCl-H₂O system at 1 – 4 GPa and 800 – 1100°C using piston cylinder apparatus, and discusses the interconnection of aqueous fluid in peridotite just above the subducted slab based on the percolation theory. The experimental results showed that dihedral angle becomes less than 60° at sufficiently lower temperature when aqueous fluid contains significant amount of NaCl. The authors propose that the released NaCl-bearing aqueous fluid can migrate at depth less than 80 km, which is not consistent with a position of volcanic front. Finally they conclude that the release depth of aqueous fluid is not a primary control on the volcanic front position against the model proposed by Mibe et al. (1999). However, I feel that this conclusion is not obvious if we consider error of dihedral angle measurement and thermal structure of the mantle wedge inferred from the latest models. Rather this paper should make more effort to explain origin of high electrical conductivity anomalies at a nearly constant

distance of 20-40 km seaward from the volcanic front. Although the paper contains interesting issues, I do not feel the large impact enough to publish in Nature Geoscience.

General comments

A main discussion point of this paper is accurate determination of the critical boundary for isolation and percolation in P-T space. In an ideal isotropic system, interconnection of aqueous fluid allowing permeable flow is completely controlled by dihedral angle. However, the measured angles are close to 60° (within 10° in most case), which is a critical angle for percolation. Although they have mentioned that measurement error is 3° , we know that the measurement accuracy is strongly controlled by criteria to define dihedral angle. For example, grain boundary seems to be opened as shown in Fig. 1c. How to define the dihedral angle in this case? How to eliminate data of dihedral angle containing faceted surface? I agree with relative reduction of dihedral angle by addition of NaCl at low temperature (800°C). Given the uncertainties of dihedral angle measurement, the argument against percolation, however, seems to be not so obvious.

The location of the critical boundary for percolation is governed by thermal structure of mantle wedge. Which thermal model was used in Fig. 7? This thermal structure is largely different from those predicted from the latest thermal models. According to recent thermal models by England and Katz (2010), van Keken et al. (2011) and Perrin et al. (2018), the slab surface temperature abruptly increases from 400 to 1200°C at a constant depth around 80 km. As a result, depth interval for establishment of interconnection between pure H₂O and NaCl-bearing is quite narrow. This narrow depth interval make it difficult to discuss the volcanic front position due to the difference of fluid composition. The authors should discuss position of the critical boundary for interconnection using thermal structures obtained from the recent papers.

Minor comments

Fig. 1: Please provide an example showing lines to determine dihedral angle. Measurement of dihedral angle is generally acquired by image analysis of secondary electron images because surface morphology is much clear than the back-scattered electron image. Why did the authors use the back scattered image?

Fig. 2: Please provide one figure showing difference with or without dihedral angle including faceted surface.

Line 175-208: Cmíral et al. (1998) suggested that the measured dihedral angle tends to decrease with increasing resolution of image. In this study, they have obtained high resolution image using field emission SEM. I am amazed that the results are almost consistent.

Line 187-188: Does proportion of faceted surface increase with decreasing temperature?

Line 201-208: I do not think it makes sense to do the same experiment in different laboratory to confirm the reproducibility of the experiment.

Line 219-220: Was it not published as paper?

Line 266-268: Are there no conductivity anomalies in these regions? Conductivity anomaly was found in Cascadia. It seems to be difficult to consider fluid as a cause of electric conductivity anomaly observed in subduction zones in a unified way. Incidentally, electrical conductivity of serpentinite is not high enough to explain the observed conductivity (Bruno et al., 2011 EPSL; Guo et al., 2012 PEPI).

Line 291-293: If the system contains NaCl, the water activity is not unity. Dehydration temperature of brine-bearing altered oceanic crust may be higher than they thought. Which mineral phase in hydrated MORB is the main container for NaCl?

[Response to each comment]

We deeply thank both reviewers for their thoughtful reviews and constructive comments.

We revised all of the points according to the reviewers' comments.

Reviewer#1

Main comments referring to line Nr in paper:

1) Lines 4, 29-30, 36-37 suggest that a hidden reservoir is needed to balance the inputs and outputs in subduction zones. This is in conflict to the correct statement in lines 22-24, that there is a continuous release of water from the subducted crust during prograde subduction metamorphism. We are far away from fully quantifying how much water is recycled from the slab via fore-arc mantle or accretionary prisms to the crust. It seems a problem is created here to offer a solution that might not be needed.

We have revised the manuscript to limit our implication for the missing water issue in subduction zones (Lines 16, Lines 61–62).

2) Lines 53-54. The dihedral angle of the olivine fluid-system is controlling fluid flow at low fluid fractions. All experiments were performed at low fluid contents of 3-5 wt%. However, it is not clear if this is the case when a fluid enters the mantle wedge. If fluids are collected in shear zones in the subducting slabs, it is very likely that the fluid fraction is high and in such case the fluid is expected to be mobile. The generalisations are not always justified.

We have mentioned a possible occurrence of high fluid fraction in shear zones in which the critical dihedral angle can be larger than 60° (Lines 73–76).

3) Lines 62-70: This is quite a good summary of evidence for saline fluids in subduction zones. The very high NaCl contents in Alpine serpentinite have to be taken with care as they most likely derive from a desiccation process (see also point). Overall most evidence points to salinities < 5 wt% NaCl. The main investigated conditions with 27.5 % NaCl is really on the high side for aqueous subduction zone fluids. I acknowledge that two salinity series were done at constant conditions, indicating that the results are also valid at lower salinities, but it would have been nice to have the bulk of the data at more realistic salinities.

We agree with the reviewer's comment. We deleted the high salinity example of Alpine serpentinite. Then, we conducted additional experiments at more realistic salinities along with systematic survey of the saline concentration dependence. The new experiments include 1) 3.0 wt% and 5.0 wt% NaCl at 1 GPa and 800 °C; 2) 1.0 wt%, 3.0 wt%, and 10.0 wt% NaCl at 2 GPa and 800 °C; 3) 1.0 wt%, 3.0 wt%, and 10.0 wt% NaCl at 2 GPa and 1000 °C; and 4) 3.0 wt% and 5.0 wt% NaCl at 3 GPa and 800 °C. The new results are shown in Fig. 4 and in Lines 25–26, 187–200. Importantly, the dihedral angle became lower than 60 even at 1.0 wt% NaCl.

4) 215-216 The position of the second critical point is highly debated and the conditions at 3.8 GPa, 1000°C refer to a fertile peridotite+H₂O system. However, in the current study deals with brine-olivine and the second critical point for this system is nowhere near the investigated P-T conditions but at much higher pressures and temperatures (constrained by the wet melting curve of Mg-rich olivine). Thus, this argument cannot be taken to explain the experimental results.

We agree with the reviewer's comments. We deleted the sentences concerning the second critical point in the system of olivine and NaCl-bearing aqueous fluids.

5) 249-252 Fluid production in subducted mafic crust is more complex and most eclogites are dry at 650-700°C, 25 kbar. This doesn't really fit the model as at these conditions, the dihedral angle is still > 60° even for the saline fluids. Thus, fluid transport in the mantle wedge according to the presented

arguments is not likely and much of this fluid might escape along the slab interface. Alternatively, especially at mantle wedge temperatures $< 650^{\circ}\text{C}$ serpentine minerals will form due to fluid infiltration. It would be good to also show the stability of antigorite in Figure 6. This would highlight that fluids derived from mafic rocks would mainly be produced in a region, where antigorite is still stable in the mantle wedge.

We appreciate this comment. According to Fig. 3 of van Keken et al. (2011), the top of subducting mafic crust initiates eclogite transformation from a depth of 80 km (assumed decoupling depth) in the intermediate-temperature and cold subduction zones. Such transformation is delayed down to the depths of ~ 150 km at deeper parts of the subducting mafic crust because of the inverted thermal gradient in the subducting slab. Therefore, continuous fluid supply from the subducting oceanic crust to the permeable mantle wedge can be expected at a depth range of 80–150 km in the intermediate-temperature and cold subduction zones. We drew the stability of antigorite and chlorite in Figs. 6 and 7 according to this comment.

6) Lines 271-77; Figure 7 284-288. As noted above, the dehydration of oceanic crust is not correctly shown in this figure, as most of the oceanic crust at the top of the slab will be anhydrous before the coupling point is reached. It is also important to note that in Figure 6 there is a gap between the stability of antigorite in the mantle wedge and the line of $\Theta = 60^{\circ}$. This gap somehow disappeared in Figure 7. However, this is quite important for how fluid is transported in this critical region in the mantle wedge.

According to this comment, we revised the Figs. 6 and 7. We showed the coupling point (80 km; van Keken et al., 2011; Syracuse et al., 2010; Wada and Wang, 2009), and we deleted the detailed structure in the subducting slab in Fig. 7. Moreover, we drew the stability of antigorite in Figs. 6 and 7. The gap between the antigorite stability and dihedral angle threshold is now shown in Fig. 7.

7) 291-294. Here it is assumed that the saline fluid is produced directly in the subducted slab. This is not easy to do. The Cl/H₂O ratio of dehydrating phases is generally much lower than salinities of 5 wt% (see for example Li and Hermann, 2015). One proposed mechanism that has been used to also explain the electric conductivity is that saline fluids are not derived from the slab but are generated by the formation of hydrous phases such as serpentine and chlorite in the mantle wedge where fluids are getting enriched in Cl by desiccation (see for example Reynard et al., 2011, EPSL)

According to this comment, we cited the suggested references and discussed the evolution of salinity of the slab-derived aqueous fluid (Lines 90–96, 356–358).

8) Throughout the discussion (280-284), the authors give the impression that arc lavas are just formed by the addition of water to the mantle wedge. While water is the key ingredient, all arc lavas are also characterised by a “slab component” that contains major and trace elements. With the proposed model, the main subduction fluids are released from the slab at quite low temperatures and pressures and then they travel to hotter parts trapped within the down-dragged coupled mantle. At 500-600°C, even in highly saline fluids, the solubility of LREE is far too low to explain the H₂O/Ce ratios in arc lavas (see for example Plank et al., 2009). Thus, the proposed model might partially work for water but it seems difficult to reconcile it with additional constraints on the composition of subduction zone fluids.

According to this comment, we included the release of water from the down-dragged coupled mantle in Figs. 7a and 7b. We also deleted the statements that suggest direct and voluminous addition of water by eclogite transformation in the subducting slab to the mantle wedge. Regarding the trace element transport to arc magmas, we briefly mentioned a possible contribution of hydrous silicate melt of the subducting slab produced at deep depths. However, we shortened the discussion on the chemistry (Lines 361–367) and focused on the formation of the high electrical conductivity region because our emphasis is on the networking and mode of migration of saline aqueous fluid in the

mantle wedge, particularly in the fore-arc region (Lines 284–291, 345–352). In addition, the hydrous peridotite solidus was revised based on more recent research (Lines 339–345, Figs. 6 and 7).

Additional General comments:

9) Only static conditions at low fluid fractions are investigated. However, the subduction channel and mantle wedge are dynamic environments and thus fluid flow in a deforming system might be quite different. This point needs to be addressed.

We agree with this comment. In the revised manuscript, we added sentences regarding similarities in the pore structure and dihedral angle in a solid–liquid system between deformed and static experimental products (Lines 292–295).

10) It has been proposed based on different interaction features of slab fluids with the olivine that fluid flow in the mantle wedge might be channelized (see for example Pirard and Hermann, 2015 Geology).

We agree that fluid focusing or other self-organization could occur by interaction between the fluids and the solid matrix; however, the dihedral angle is still important for controlling the fluid connectivity. Pirard and Hermann (2015) investigated the slab-derived hydrous melt at 3.5 GPa at a depth of about 100 km, where the slab temperature can be higher than the slab wet solidus. However, in our study we focused mainly on relatively shallow depths of about 80 km, where the slab-released fluid is mainly aqueous fluid rather than hydrous melt (Lines 363–367).

11) Porosity waves as advocated by Connolly and Podladchikov are not considered as a potential mechanism to transport fluids through the mantle wedge. This must be addressed as well.

According to this comment, we cited new references and revised the manuscript to address porosity waves in the mantle wedge (Lines 307–314). Importantly, a permeable mantle wedge ($\theta < 60^\circ$) is required for the occurrence of porosity waves.

12) I recommend limitation of the application of the results to the conductivity of fluids within a more static low T mantle wedge, rather than trying to explain the whole subduction-recycling system. Especially in the region where chlorite-peridotite is stable in the fore-arc region, the model could be nicely applied as the formation of chlorite would provide a means to create high salinity fluids, and the dihedral angle with olivine could explain then the migration of these fluids through the fore-arc, providing an explanation for the electrical conductivity and magnetotelluric studies.

We agree with reviewer's comment. We limited our implication and revised our model in Fig. 7. We focused mainly on the migration of slab-derived saline fluids at the fore-arc mantle wedge and emphasized the origin of high electrical conductivity beneath the fore-arc region (Lines 28–31, 263–264, 316). In addition, we drew the stability of chlorite in Fig. 7.

Figure 6 is very schematic and hard to read. The depth temperature paths of the subducted slabs are very systematic. It is highly unlikely that all subduction zones will couple to the mantle at exactly the same depth. It would be good to show also the antigorite stability in this diagram. The wet peridotite solidus is not accurate as well, especially at the low pressure and high pressure end of the solidus.

We added the antigorite stability in Fig. 6. The depth–temperature paths of the subducted slabs in Fig. 6 are cited from the model (Syracuse et al., 2010). Wada and Wang (2009) concluded that the depth to decoupling is uniform at depth of ~80 km, which indicates a universal mechanism governing the decoupling between the slab and the mantle. We cited a new reference, Till et al. (2012), to show the precise hydrous peridotite solidus.

Figure 7 does only partly reflect the data in Figure 6 (see point 6).

We revised Figs. 6 and 7 to ensure that they are consistent with each other.

Reviewer #2 (Remarks to the Author):

The paper reports a variation of dihedral angle of olivine-NaCl-H₂O system at 1 – 4 GPa and 800 – 1100°C using piston cylinder apparatus, and discusses the interconnection of aqueous fluid in peridotite just above the subducted slab based on the percolation theory. The experimental results showed that dihedral angle becomes less than 60° at sufficiently lower temperature when aqueous fluid contains significant amount of NaCl. The authors propose that the released NaCl-bearing aqueous fluid can migrate at depth less than 80 km, which is not consistent with a position of volcanic front. Finally, they conclude that the release depth of aqueous fluid is not a primary control on the volcanic front position against the model proposed by Mibe et al. (1999). However, I feel that this conclusion is not obvious if we consider error of dihedral angle measurement and thermal structure of the mantle wedge inferred from the latest models. Rather this paper should make more effort to explain origin of high electrical conductivity anomalies at a nearly constant distance of 20-40 km seaward from the volcanic front. Although the paper contains interesting issues, I do not feel the large impact enough to publish in Nature Geoscience.

General comments

A main discussion point of this paper is accurate determination of the critical boundary for isolation and percolation in P-T space. In an ideal isotropic system, interconnection of aqueous fluid allowing permeable flow is completely controlled by dihedral angle. However, the measured angles are close to 60° (within 10° in most case), which is a critical angle for percolation. Although they have mentioned that measurement error is 3°, we know that the measurement accuracy is strongly controlled by criteria to define dihedral angle. For example, grain boundary seems to be opened as shown in Fig. 1c. How to define the dihedral angle in this case? How to eliminate data of dihedral angle containing faceted surface? I agree with relative reduction of dihedral angle by addition of NaCl at low temperature (800°C). Given the uncertainties of dihedral angle measurement, the argument against percolation, however, seems to be not so obvious.

We appreciate the reviewer's comment. We revised Figure 1 and added Supplementary Note 1 and Supplementary Figs. 1, 2, and 3 to show our process for selection and measurement of the dihedral angle in this study. We did not select angles having one or two flat interfaces as the measured dihedral angles. In addition, we discussed the error of the measurement in Supplementary Note 1.

The location of the critical boundary for percolation is governed by thermal structure of mantle wedge. Which thermal model was used in Fig. 7? This thermal structure is largely different from those predicted from the latest thermal models. According to recent thermal models by England and Katz (2010), van Keken et al. (2011) and Perrin et al. (2018), the slab surface temperature abruptly increases from 400 to 1200 °C at a constant depth around 80 km. As a result, depth interval for establishment of interconnection between pure H₂O and NaCl-bearing is quite narrow. This narrow depth interval makes it difficult to discuss the volcanic front position due to the difference of fluid composition. The authors should discuss position of the critical boundary for interconnection using thermal structures obtained from the recent papers.

We appreciate this comment. We modified the geothermal structure in Figure 7 based on Wada et al. (2008) to ensure that Figs. 6 and 7 are consistent with each other. We employed the current thermal model of Syracuse et al. (2010) and van Keken et al. (2011) in Fig. 6, which is essentially the same as that used in Wada et al. (2008) and in Wada and Wang (2009).

In this model, the slab surface temperature abruptly increases from 400 to 800 °C due to coupling between subducted slab and overriding mantle wedge over a few kilometers.

Minor comments

Fig. 1: Please provide an example showing lines to determine dihedral angle. Measurement of dihedral angle is generally acquired by image analysis of secondary electron images because surface morphology is much clearer than the back-scattered electron image. Why did the authors use the back-scattered image?

We appreciate this comment. In our study, we used secondary electron images for dihedral angle measurement rather than back-scattered electron images. Additional examples that show our selection and measurement process are presented in Supplementary Note 1 and in Supplementary Figs. 1–3.

Fig. 2: Please provide one figure showing difference with or without dihedral angle including faceted surface.

We appreciate this comment. We presented two typical secondary electron images that show dihedral angles with and without faceted surfaces in Supplementary Fig. 2.

Line 175-208: Cmíral et al. (1998) suggested that the measured dihedral angle tends to decrease with increasing resolution of image. In this study, they have obtained high resolution image using field emission SEM. I am amazed that the results are almost consistent.

We believe that the high-resolution and high-magnification image enables us to conduct a precise angle measurement. Such an image, however, does not always result in a lower dihedral angle. For example, the dihedral angle in the H₂O system at 1 GPa and 1000 °C in our study is consistent with the results of Watson and Brenan (1987), in which the lower magnification image was utilized for the angle measurement.

Line 187-188: Does proportion of faceted surface increase with decreasing temperature?

Although we did not measure the proportion of faceted surfaces in our experiments, Rottman and Wortis (1984) theoretically demonstrated that the proportion of faceted surfaces tends to decrease with an increase in temperature.

Line 201-208: I do not think it makes sense to do the same experiment in different laboratory to confirm the reproducibility of the experiment.

We appreciate the reviewer's comment. We tried to confirm the errors for the dihedral angle originating from the differences in experimental procedure and cell assemblies of the piston cylinder experiment. We concluded that such errors are negligible because the dihedral angle obtained at BGI and Tohoku University showed essentially the same values under the same experimental conditions.

Line 219-220: Was it not published as paper?

This was not published. We have not modified manuscript for this point because it is not necessary.

Line 266-268: Are there no conductivity anomalies in these regions? Conductivity anomaly was found in Cascadia. It seems to be difficult to consider fluid as a cause of electric conductivity anomaly observed in subduction zones in a unified way. Incidentally, electrical conductivity of serpentinite is not high enough to explain the observed conductivity (Bruno et al., 2011 EPSL; Guo et al., 2012 PEPI).

We did not change manuscript for this comment because it is not necessary.

The geophysical observations have also found high electrical conductivity anomalies in the fore-arc region of the Cascadia subduction zone (Soyer and Unsworth, 2006; Wannamaker et al., 2014). Because the eclogite transformation occurs at shallow depths of 40–60 km in the Cascadia zone, the

slab-derived fluid leaking through the highly serpentinized fore-arc mantle could be responsible for these anomalies (Reynard et al., 2011; Nakatani and Nakamura, 2016).

Line 291-293: If the system contains NaCl, the water activity is not unity. Dehydration temperature of brine-bearing altered oceanic crust may be higher than they thought. Which mineral phase in hydrated MORB is the main container for NaCl?

We did not change manuscript for this comment because it is not reasonable.

Reduction of water activity leads to lower dehydration temperature. Previous research has indicated that the dehydration of hydrous phases occurs at shallower depths in the presence of dissolved NaCl in the fluid (Mantegazzi et al., 2013). However, its effect may be limited owing to the low salinity (<~5 wt%) of slab-derived fluid equilibrated with subducting lithologies (e.g., Li and Hermann, 2015). In the subducted sediment and oceanic crust, Na can be accommodated in jadeite and paragonite, and Cl can be carried mainly by Cl-rich pargasite, apatite, and in the fluid phase and to a lesser extent by phengite and biotite.

Reference

- van Keken, Peter E., et al. Subduction factory: 4. Depth-dependent flux of H₂O from subducting slabs worldwide. *J. Geophys. Res.* **116**, B1 (2011).
- Syracuse, E. M., van Keken, P. E., Abers, G. A. The global range of subduction zone thermal models. *Phys. Earth Planet. Inter.* **183**, 73–90 (2010).
- Wada, I., Wang, K. Common depth of slab-mantle decoupling: Reconciling diversity and uniformity of subduction zones. *Geochem. Geophys. Geosyst.* **10** (2009).
- Pirard, C., Hermann, J. Focused fluid transfer through the mantle above subduction zones. *Geology* **43** (10), 915–918 (2015).
- Till, C. B., Grove, T. L., Withers, A. C. The beginnings of hydrous mantle wedge melting. *Contrib. Mineral. Petrol.* **163** (4), 669–688 (2012).
- Wada, I., Wang, K., He, J., Hyndman, R. D. Weakening of the subduction interface and its effects on surface heat flow, slab dehydration, and mantle wedge serpentinization. *J. Geophys. Res.* **113** (B4) (2008).
- Watson, E. B., Brenan, J. M. Fluids in the lithosphere, 1. Experimentally-determined wetting characteristics of CO₂–H₂O fluids and their implications for fluid transport, host-rock physical properties, and fluid inclusion formation. *Earth Planet. Sci. Lett.* **85**, 497–515 (1987).
- Rottman, C., Wortis, M. Statistical mechanics of equilibrium crystal shapes: Interfacial phase diagrams and phase transitions. *Phys. Rep.* **103** (1-4), 59-79 (1984).
- Soyer, W., Unsworth, M. Deep electrical structure of the northern Cascadia (British Columbia, Canada) subduction zone: Implications for the distribution of fluids. *Geology* **34**, 53–56 (2006).
- Wannamaker, P. E., Evans, R. L., Bedrosian, P. A., Unsworth, M. J., Maris, V., McGary, R. S. Segmentation of plate coupling, fate of subduction fluids, and modes of arc magmatism in Cascadia, inferred from magnetotelluric resistivity. *Geochem Geophys.* **15** (11), 4230-4253 (2014).
- Reynard, B., Mibe, K., Van de Moortèle, B. Electrical conductivity of the serpentinised mantle and fluid flow in subduction zones. *Earth Planet. Sci. Lett.* **307** (3–4), 387–394 (2011).

Nakatani, T., Nakamura, M. Experimental constraints on the serpentinization rate of fore-arc peridotites: Implications for the upwelling condition of the slab-derived fluid. *Geochem Geophys.* **17**, 3393–3419 (2016).

Mantegazzi, D., Sanchez-Valle, C., Driesner, T. Thermodynamic properties of aqueous NaCl solutions to 1073 K and 4.5 GPa, and implications for dehydration reactions in subducting slabs. *Geochim. Cosmochim. Acta.* **121**, 263-290 (2013).

Li, H., Hermann, J. Apatite as an indicator of fluid salinity: An experimental study of chlorine and fluorine partitioning in subducted sediments. *Geochim. Cosmochim. Acta.* **166**, 267–297 (2015).

Reviewers' Comments:

Reviewer #2:

Remarks to the Author:

The paper reports a variation of dihedral angle of olivine-NaCl-H₂O system at 1 – 4 GPa and 800 – 1100 °C using piston cylinder apparatus, and discusses the shallow fluid circulation in mantle wedge from the viewpoint of the interconnection of aqueous fluid in peridotite just above the subducted slab based on the percolation theory. First of all, I thank the authors for their effort to conduct further experiments and add a detailed note on careful handling to determine dihedral angles. The experimental results showed that dihedral angle becomes less than 60° at sufficiently lower temperature when aqueous fluid contains certain amount of NaCl even at 1 wt%. Now I am convinced that the systematic variation of dihedral angle reported in this study is sufficiently reliable data.

I feel the main point of this manuscript has changed in this revision. The original version emphasized that the release depth of aqueous fluid is not a primary control on the volcanic front position against the model proposed by Mibe et al. (1999). In this revision, the main arguments focused on the origin of high electrical conductivity anomalies at ~80 km depth and a nearly constant distance of 20-40 km seaward from the volcanic front. However, assessment of fluid connectivity determined in this study is not directly utilized to account for the conductivity anomalies observed in various subduction zones. Although interconnection of fluid phase in rock is generally considered as an indicator to raise electrical conductivity, no such argument has been made by comparison with geophysical observations in the results of this study.

Considering steep thermal gradient at the slab surface at depth of around 80 km, the depth difference between serpentine dehydration and establishment of interconnection of saline-fluid is very narrow. On the other hand, the conductivity anomalies can be generated by antigorite dehydration process irrespective of salinity because dehydration of antigorite can serve significant amount of water. In such a case, the movement of the fluid does not follow the percolation theory applied to systems with low fluid fraction. Therefore, it is difficult to distinguish between two effects as an origin of conductivity anomalies observed at 80 km depth. In addition, it is well known that the seismicity observed in the subduction zone is closely related to the dehydration of hydrous minerals. Thus hydraulic fracturing has been considered to be a dominant fluid transport mechanism in this area. The released water can be effectively drained by fracture network without trapping of water at grain corners or along grain boundary. These views do not necessarily support their interpretation that connectivity of saline fluid controls the electrical conductivity anomalies observed at the subduction zones.

The paper reports a variation of dihedral angle of olivine-NaCl-H₂O system at 1 – 4 GPa and 800 – 1100 °C using piston cylinder apparatus, and discusses the shallow fluid circulation in mantle wedge from the viewpoint of the interconnection of aqueous fluid in peridotite just above the subducted slab based on the percolation theory. First of all, I thank the authors for their effort to conduct further experiments and add a detailed note on careful handling to determine dihedral angles. The experimental results showed that dihedral angle becomes less than 60° at sufficiently lower temperature when aqueous fluid contains certain amount of NaCl even at 1 wt%. Now I am convinced that the systematic variation of dihedral angle reported in this study is sufficiently reliable data.

I feel the main point of this manuscript has changed in this revision. The original version emphasized that the release depth of aqueous fluid is not a primary control on the volcanic front position against the model proposed by Mibe et al. (1999). In this revision, the main arguments focused on the origin of high electrical conductivity anomalies at ~80 km depth and a nearly constant distance of 20-40 km seaward from the volcanic front. They propose a new idea that a "permeable window" of mantle wedge exists for migration of slab derived saline fluid. However, I feel that assessment of fluid connectivity determined in this study is not directly utilized to account for the conductivity anomalies observed in various subduction zones. Although interconnection of fluid phase in rock is generally considered as an indicator to raise electrical conductivity, no such

argument has been made by comparison with geophysical observations in the results of this study. Indeed, there is no correlation between where the conductivity should be high due to the establishment of fluid interconnection and where the conductivity obtained from observation is high.

General comments

Considering steep thermal gradient at the slab surface at depth of around 80 km, the depth difference between serpentine dehydration and establishment of interconnection of saline-fluid is very narrow. On the other hand, the conductivity anomalies can be generated by antigorite dehydration process irrespective of salinity because dehydration of antigorite can serve significant amount of water. In such a case, the movement of the fluid does not follow the percolation theory applied to systems with low fluid fraction. Therefore, it is difficult to distinguish between two effects as an origin of conductivity anomalies observed at 80 km depth. In addition, it is well known that the seismicity observed in the subduction zone is closely related to the dehydration of hydrous minerals. Thus hydraulic fracturing has been considered to be a dominant fluid transport mechanism in this area. The released water can be effectively drained by fracture network without trapping of water at grain corners or along grain boundary. These views do not necessarily support their interpretation that connectivity of saline fluid controls the electrical conductivity anomalies observed at the subduction zones.

An accumulation of fluids at ~20–30 km depth at a distance of 30 km seaward from the volcanic arc has been observed as a global phenomenon. The main discussion in this paper should make more efforts to explain the observed conductivity anomalies in the shallow part with the penetration of saline fluid. I still cannot be convinced that the change of wetting properties made it possible to supply fluid to the high conductive region at ~20–30 km depth. If the fluid moves upwards steadily by permeable flow, high conductivity anomalies should be observed throughout the fore-arc area where interconnection of saline fluid is established. However, such features have not been recognized in the mantle wedge region from electromagnetic observations, rather the conductivity anomalies are observed locally. The mechanism of hydraulic fracturing allows the upward movement of fluid in the cold region on the forearc side of the mantle wedge because of high density contrast between rock and fluid. It seems that intermittent fluid movement by fracturing process can explain the observation more reasonably, as this mechanism does not create steadily a conductivity anomaly in the area. High conductive anomalies observed at crustal depth is in the region where the dihedral angle is over 60°. These facts mean that the degree of connectivity and the conductivity anomaly are irrelevant. In summary, there is no robust evidence that wetting behavior of olivine coexisting with saline fluid controls the fluid circulation in mantle wedge.

Reviewer #3:

Remarks to the Author:

Dear authors,

You did a very thorough revision and explained in detail what you have changed. I especially appreciate the additional experiments at low salinities. These additional experiments are essential for a wide application of the results and they address my main point of criticism of the earlier version.

I am satisfied with the revision and explanations and I would like to congratulate you for a very original and interesting contribution that will attract the attention of a wide range of researchers dealing with subduction zones. Therefore, I recommend that the paper should be published in Nature Communications. I only have some very minor points. Comments refer to line number in the manuscript.

240 is indicative of reduced

342 The position of the wet solidus is debated and the results of Till et al. have been questioned in recent times (see for example Green, 2015; *Phy Chem Minerals* for a review). As this is not a key point of the paper, I suggest to play it safe and also cite the alternative position of the wet solidus at about 1050°C at these depths.

353 2-3 GPa (space missing).

Bern 8.9.19

Jörg Hermann

[Response to each comment]

We deeply thank both reviewers for their thoughtful reviews and constructive comments.

We revised all of the points according to the reviewers' comments.

Reviewers' comments:

Reviewer #2 (Remarks to the Author).

The paper reports a variation of dihedral angle of olivine-NaCl-H₂O system at 1–4 GPa and 800–1100 °C using piston cylinder apparatus, and discusses the shallow fluid circulation in mantle wedge from the viewpoint of the interconnection of aqueous fluid in peridotite just above the subducted slab based on the percolation theory. First of all, I thank the authors for their effort to conduct further experiments and add a detailed note on careful handling to determine dihedral angles. The experimental results showed that dihedral angle becomes less than 60° at sufficiently lower temperature when aqueous fluid contains certain amount of NaCl even at 1 wt%. Now I am convinced that the systematic variation of dihedral angle reported in this study is sufficiently reliable data.

I feel the main point of this manuscript has changed in this revision. The original version emphasized that the release depth of aqueous fluid is not a primary control on the volcanic front position against the model proposed by Mibe et al. (1999). In this revision, the main arguments focused on the origin of high electrical conductivity anomalies at ~80 km depth and a nearly constant distance of 20-40 km seaward from the volcanic front. They propose a new idea that a “permeable window” of mantle wedge exists for migration of slab derived saline fluid.

1. However, I feel that assessment of fluid connectivity determined in this study is not directly utilized to account for the conductivity anomalies observed in various subduction zones. Although interconnection of fluid phase in rock is generally considered as an indicator to raise electrical conductivity, no such argument has been made by comparison with geophysical observations in the results of this study. Indeed, there is no correlation between where the conductivity should be high due to the establishment of fluid interconnection and where the conductivity obtained from observation is high.

We have cited the most recent references which have shown almost continuous anomalies of electrical conductivity from the forearc mantle to the lower crust in a forearc region (Worzewski et al., 2011; Pommier and Evans, 2017; Araya et al., 2019). We have pointed out that the fluid-fraction dependence of the conductivity (Shimajuku et al., 2014) and the electrical static effect of the shallow high electrical conductor on electrical amplitude response of the deep electrical field (Torres-Verdin and Bostick, 1992). We have stated that the electrical conductivity can be enhanced by the high fluid fraction due to the fluid accumulation over the geological time scale and high fluid salinity resulted from the hydration of the mantle wedge during transport. We have also emphasized that the aqueous fluid phase is the only plausible candidate to account for high electrical conductivity anomalies in forearc regions. We have revised our manuscript in **Lines 339–352**.

2. Considering steep thermal gradient at the slab surface at depth of around 80 km, the depth difference between serpentine dehydration and establishment of interconnection of saline-fluid is very narrow. On the other hand, the conductivity anomalies can be generated by antigorite dehydration process irrespective of salinity because dehydration of antigorite can serve significant amount of water. In such a case, the movement of the fluid does not follow the percolation theory applied to systems with low fluid fraction. Therefore, it is difficult to distinguish between two effects as an origin of conductivity anomalies observed at 80 km depth. In addition, it is well known that the seismicity observed in the subduction zone is closely related to the dehydration of hydrous minerals. Thus, hydraulic fracturing has been considered to be a dominant fluid transport mechanism in this area. The released water can be effectively drained by fracture network without trapping of water at grain corners or along grain boundary. These views do not necessarily support their interpretation that connectivity of saline fluid controls the electrical conductivity anomalies observed at the subduction zones.

Thanks a lot for this comment, but the reviewer seems to misunderstand our point.

Our study focuses on the fluid circulation in the forearc region, which can provide an interpretation of high electrical conductivity anomalies. The aqueous fluids can be produced by dehydration of a sinking slab or down-dragged basal layer of the mantle. The point is that the “permeable window” in our model exists between the serpentinite stable field (cold and stagnant wedge core) and the partial melting setting (hot wedge core), which makes possible the infiltration of aqueous fluids through forearc mantle to lower crust because of the small dihedral angle in the olivine-H₂O-NaCl system. We don’t deny the possibility of hydraulic fracturing, but it is not necessary if we consider the recent magnetotelluric results and electrical static effect as described in Lines 339–352.

3. An accumulation of fluids at ~20–30 km depth at a distance of 30 km seaward from the volcanic arc has been observed as a global phenomenon. The main discussion in this paper should make more efforts to explain the observed conductivity anomalies in the shallow part with the penetration of saline fluid. I still cannot be convinced that the change of wetting properties made it possible to supply fluid to the high conductive region at ~20–30 km depth. If the fluid moves upwards steadily by permeable flow, high conductivity anomalies should be observed throughout the fore-arc area where interconnection of saline fluid is established. However, such features have not been recognized in the mantle wedge region from electromagnetic observations, rather the conductivity anomalies are observed locally. The mechanism of hydraulic fracturing allows the upward movement of fluid in the cold region on the forearc side of the mantle wedge because of high density contrast between rock and fluid. It seems that intermittent fluid movement by fracturing process can explain the observation more reasonably, as this mechanism does not create steadily a conductivity anomaly in the area. High conductive anomalies observed at crustal depth is in the region where the dihedral angle is over 60°. These facts mean that the degree of connectivity and the conductivity anomaly are irrelevant. In summary, there is no robust evidence that wetting behavior of olivine coexisting with saline fluid controls the fluid circulation in mantle wedge.

This comment is almost the same as the comments 1 and 2, to which we have already replied. The highest electrical conductivity anomalies in the crust is now widely proposed as a consequence of the accumulation of aqueous fluids. Obviously, such a fluid reservoir in the lower crust needs a supply of fluids. Our steady-state fluid circulation model provides an answer to this issue.

Reviewer #3 (Remarks to the Author):

Dear authors,

You did a very thorough revision and explained in detail what you have changed. I especially appreciate the additional experiments at low salinities. These additional experiments are essential for a wide application of the results and they address my main point of criticism of the earlier version.

I am satisfied with the revision and explanations and I would like to congratulate you for a very original and interesting contribution that will attract the attention of a wide range of researchers dealing with subduction zones. Therefore, I recommend that the paper should be published in Nature Communications. I only have some very minor points. Comments refer to line number in the manuscript.

240 is indicative of reduced

342 The position of the wet solidus is debated and the results of Till et al. have been questioned in recent times (see for example Green, 2015; Phy Chem Minerals for a review). As this is not a key point of the paper, I suggest to play it safe and also cite the alternative position of the wet solidus at about 1050°C at these depths.

353 2-3 GPa (space missing).

We have modified all points following the reviewer's suggestions in Lines 240, 355–360, 371 and Figures 6, 7, and Supplementary Figures 7–9.

Bern
8.9.19
Jörg Hermann

Reviewers' Comments:

Reviewer #2:

Remarks to the Author:

This paper investigated effect of salinity on wetting behavior of olivine aggregates well. This paper gives an interesting result based on high pressure experiments to understand fore-arc conductors observed in mantle wedge regions. Now I am satisfied with the revision and explanations. Their interpretation on fluid flow pattern in the mantle wedge region controlled by wetting behavior seems to be supported by electrical conductivity structures of some mantle wedge regions obtained from magnetotelluric studies. As I pointed out before, fluid flow occurs not only by wetting properties of rocks but also by hydraulic fracturing. The location of fore-arc conductor can be explained by a change of wetting behavior from isolation to interconnection in the mantle wedge just above the subducting slab, if fluid flow occurs vertically (direction of gravity).

I think it is worthy for readers to mention the following points to keep in mind in the revised manuscript. Chlorite has a wide stability range in the P-T region of interest for this study. I think a released water from the slab is largely consumed by chlorite formation in the mantle wedge, suggesting that sustainable supply of fluid is required to maintain water saturated condition. Chlorite is likely to form along grain boundaries between nominally anhydrous minerals such as olivine, orthopyroxene, clinopyroxene and spinel. Thus, the wetting behavior seems to be rather controlled by wetting behavior of chlorite and other surrounding minerals. The wetting behavior of hydrous minerals is not same as that of olivine, and is strongly influenced by faceting of mineral surfaces. The other possible interpretation allowing their thoughts is that the released water at the beginning of dehydration has relatively higher salinity. Because the released aqueous fluid with high salinity can reduce the water activity effectively, stability limit of chlorite would shift to lower temperature. Thus chlorite can be unstable in the area considered by this study. In this case, grain boundaries in peridotite is dominantly composed of olivine grain boundaries. The authors should mention the above issue in the revised manuscript.

I recommend that the paper should be published in Nature Communications after addressing the above comments.

[Response to each comment]

We deeply thank reviewer for his/her thoughtful reviews and constructive comments.

We revised all of the points according to the reviewers' comments.

Reviewers' comments:

Reviewer #2 (Remarks to the Author).

This paper investigated effect of salinity on wetting behavior of olivine aggregates well. This paper gives an interesting result based on high pressure experiments to understand fore-arc conductors observed in mantle wedge regions. Now I am satisfied with the revision and explanations. Their interpretation on fluid flow pattern in the mantle wedge region controlled by wetting behavior seems to be supported by electrical conductivity structures of some mantle wedge regions obtained from magnetotelluric studies. As I pointed out before, fluid flow occurs not only by wetting properties of rocks but also by hydraulic fracturing. The location of fore-arc conductor can be explained by a change of wetting behavior from isolation to interconnection in the mantle wedge just above the subducting slab, if fluid flow occurs vertically (direction of gravity).

I think it is worthy for readers to mention the following points to keep in mind in the revised manuscript. Chlorite has a wide stability range in the P-T region of interest for this study. I think a released water from the slab is largely consumed by chlorite formation in the mantle wedge, suggesting that sustainable supply of fluid is required to maintain water saturated condition. Chlorite is likely to form along grain boundaries between nominally anhydrous minerals such as olivine, orthopyroxene, clinopyroxene and spinel. Thus, the wetting behavior seems to be rather controlled by wetting behavior of chlorite and other surrounding minerals. The wetting behavior of hydrous minerals is not same as that of olivine, and is strongly influenced by faceting of mineral surfaces. The other possible interpretation allowing their thoughts is that the released water at the beginning of dehydration has relatively higher salinity. Because the released aqueous fluid with high salinity can reduce the water activity effectively, stability limit of chlorite would shift to lower temperature. Thus, chlorite can be unstable in the area considered by this study. In this case, grain boundaries in peridotite is dominantly composed of olivine grain boundaries. The authors should mention the above issue in the revised manuscript.

I recommend that the paper should be published in Nature Communications after addressing the above comments.

Thanks a lot for reviewer's comments. We have inserted the above discussions in Lines 313–321, 324–330.